# Assessing economic impacts of environmental research infrastructures: overview of methodological tools

Régis Kalaydjian[1]

[1]Ifremer, Issy-les-Moulineaux, 92130, France

This article is a summary of a research conducted within the ENVRIPLUS/H2020 project, funded by the European Commission.

*Correspondence to*: Régis Kalaydjian (regis.kalaydjian@ifremer.fr)

**Abstract.** The data generated by environmental research infrastructures (ENV RIs) are key material to analysing the quality of general living standards and the conditions of development of environmentally sensitive economic activities: monitoring
the atmosphere and ocean is increasingly and critically important in a context marked by the risks of global warming.

The primary objective of this paper is to review the main tools used to assess the economic impacts of ENV RIs and proposes a methodological framework. We have considered three impact categories: 1) upstream impacts on equipment suppliers; 2) downstream impacts on the performance and quality of observational data, monitoring services and forecasts; 3) feedback impacts in terms of improved knowledge on the environment to the benefits of economic activities. This papers'
specificity lies in the fact that the entire data and service supply chain is considered for the assessment of impacts.

An ocean-related case study is used as a practical example: Argo, a global in-situ ocean observing system, provides understanding on the supply chain from upstream infrastructure and maintenance suppliers to primary and processed ocean data providers.

The paper gives precedence to tried and tested methods. It concludes that further work and more data are needed to improve
the methods of each category.

**Keywords**: Argo; economy; environment; forecast; impact; infrastructure; monitoring; observation; ocean; research.

## 1 Introduction: objectives and impacts of ENV RIs

Research Infrastructures (RIs) are widely recognized as key components in performing high level research in many fields. In
broad terms, according to the definitions provided by the Organisation for Economic Co-operation and Development (OECD, 2019) and the European Strategy Forum for RIs (ESFRI, 2018), RIs can be defined as facilities, scientific equipment (i.e. sets of instruments, archives and scientific data, and "e-infrastructures" such as computing systems) and other tools required for research and innovation. The two reports highlight the key role of RIs in generating scientific

knowledge and in knowledge transfer through higher education and training for researchers, technology developers and innovators.

The development of RIs has become essential for governments' research policies and for international strategy aiming to co-ordinate these policies: on an international level, the Megascience Forum created by the OECD in the early 90s, and later renamed the Global Science Forum (GSF), conducts studies and produces findings and recommendations for governmental actions and international co-operation between its member states in selected fields of which RIs are a major topic. At the level of the European Union (EU), ESFRI was created in 2002 following a recommendation of the EU Council to the European Commission (EC); according to its mandate, ESFRI carries out a number of tasks and responsibilities, including: the development of an approach to policy-making on RIs in Europe, facilitating initiatives leading to a better use and development of RIs, and establishing a Roadmap for RIs for the coming 10-20 years, to be updated as and when necessary.

Despite differences between the objectives of GSF and ESFRI (the latter is specifically dedicated to European RIs), the creation of these two bodies shows the increasing importance of international organizations to RIs in terms of scientific co-operation. However the significant financial and human resources involved in developing RIs explain the need to evaluate the return on investment of these infrastructures and inform RI managers and funders of the results: OECD and ESFRI highlight the need to demonstrate the value of RIs by assessing their economic and social impacts as mentioned in the two reports referenced above; this task is part of the ESFRI Roadmap, and the OECD report offers a set of indicators, to assess the impacts of RIs' activities, including scientific, technological and educational impacts.

The present article focuses on Environmental RIs (ENV RIs), an important category of RIs considered by the ESFRI Roadmap and proposes a methodological framework to assess their economic impacts. This approach is motivated by the urgent issues emphasized by the ESFRI Roadmap and the need to respond to natural and anthropic environmental changes. The report stresses that the effect of pollution and climate change needs "to be fully understood urgently", as natural hazards "can impact society with large economic deficits". The report notes that ENV RIs address these issues through general-purpose actions (e.g. supporting education and training; delivering essential data to the public; producing accurate scientific and technical knowledge), and through specific-purpose actions (e.g. generating coherent time series of key environmental variables; opening access to environmental big data to help promote activities in the private sector; and developing new technologies for sensors, data collection and transfer).

The framework considered hereafter presents an overview of the main impact assessment methods, and an ocean-related case study, the Argo observing system. It classifies the impacts considered herein into three categories:

- Upstream impacts on equipment and infrastructure supply and maintenance services;
- Downstream impacts of ENV RIs on the performance of environmental data and monitoring services;
- Feedback impacts of observations, monitoring and improved forecast services on environmentally sensitive activities in terms of increasing efficiency and risk avoidance.

This classification is motivated by the ESFRI Roadmap and its remarks on the development of new technologies, the provision of long-term data to support downstream environmental services, and the urgent need to address major environmental challenges such as climate change.

Upstream impacts (Figure 1) concern the commercial relationships between infrastructure and equipment suppliers and maintenance service providers and ENV RI developers and operators. Suppliers' performance is boosted by a demand effect from ENV RIs. The economic impacts on suppliers are commonly assessed in terms of turnover, employment, workforce earnings, innovation and exports.

Downstream impacts (Figure 2) are generated by the uses of ENV RI data products. The performance (quality) of observational data improves that of the processed data and the information provided to data users. Certain ENV RIs develop databases for archival, scientific research and education, and stakeholder information, e.g. ICOS Integrated Carbon Observation System (www.icos-ri.eu). Other ENV RIs produce "multi-purpose" observational data (1) for archival, research and education, (2) for the development of models processing the data for environment monitoring and forecasting. (3) Further downstream, the latter datasets can be re-processed for the supply of customized data products and value-added services. The Argo case study, described below, belongs to such category, which allows the framework to include a wide spectrum of assessment methods. The methods examined in this category will give attention to data performance indicators at different stages of the data supply chain.

Feedback impacts concern the response to environmentally sensitive activities' and stakeholders' need for information on environmental conditions and changes (e.g. climate change). Comprehensive observations and a joint effort from ENV RIs are critical for understanding and predicting environmental changes. These impacts involve ENV RIs and the supply chain from upstream equipment suppliers to the diversity of downstream value-added services (Figure 3) aiming to improve scientific knowledge and so contribute to a more efficient management of environmentally sensitive activities. The methodological framework will give attention to the predictive capacity of ENV RIs. ENV RIs' contribution to risk avoidance has impacts in terms of avoided costs for a set of environmentally sensitive activities. The assessment of such avoided costs has often justified the use of cost-benefit analyses, as will be seen below.

## 2 A case study: Argo

The case study is an ocean observing system with benefits from new platform and sensor technologies and robotic measurements, but also global telecommunications. It has decisive impacts on our knowledge of oceans in terms of physics, biochemistry and linkages with climate variables. The development of Argo is therefore tightly associated to a range of upstream and downstream activities.

**2.1 Argo's role in the ocean data acquisition chain**

**2.1.1 Observation instruments and parameters**

Argo (Figure 4) is an in-situ ocean observing system providing real-time and delayed mode observations at global scale (Euro-Argo, 2018). It operates a growing array of free-drifting profiling floats measuring marine parameters on a periodical basis; the deployments began in 2000. The Core component of Argo is designed for monitoring the key ocean state variables (pressure, temperature and salinity) from the upper 2,000 m of the ocean. Recent advances in platform and sensor technologies permit to extend Core Argo's spatial coverage and develop two other key components of the programme: (1) the Biogeochemical (BGC)-Argo floats measure a range of parameters including dissolved oxygen, chlorophyll a, suspended particles, downwelling irradiance, nitrates and pH (Roemmich et al., 2019): BGC-Argo will allow better understanding of ocean biochemistry and marine resources. (2) Deep Argo is a new generation of floats designed to sample the ocean beyond the limit of 2,000 m, and capable of reaching 6,000 m, aiming to sample the full ocean depth. Deep Argo is in its pilot phase. Once deployed, it will help to better understand, inter alia, ocean warming and circulation.

The initial objective of 3,000 profiling floats was met in November 2007. The current number of floats increases by around 150-200 per year. Euro-Argo is Europe's contribution to Argo with around 600 active floats out of a total of nearly 4,000. Each float has a lifespan of 4 to 5 years.

The Argo programme brings a major contribution to the Global Ocean Observing System (GOOS) and the Global Climate Observing System (GCOS) (Riser et al., 2016; Roemmich et al., 2019), and is now key to operational oceanography and climate research. But it cannot be analysed as a separate entity; it works in association with:

- Satellite remote sensing, used to acquire global observations on altimetry, sea surface temperature and salinity, and ocean colour;

- Other in-situ observation platforms and profilers: ship based Conductivity-Temperature-Depth (CTD) profilers, expendable bathythermographs (XBTs), for temperature and depth, FerryBox systems for physical, chemical and biological parameters, moorings for a range of parameters including current, animal-mounted instruments, gliders, drifting buoys and cabled systems (a North American project is aimed at associating ocean sensors to submarine telecommunications cables - SMART subsea cables - as a complement to existing satellite and in-situ observing systems).

- In addition, Argo floats transmit observation data to satellites during the time slot when they are at the surface.

The synergy between in-situ platforms and satellites is essential:

- Platforms measure certain types of parameters that satellite profilers do not capture, e.g. sea subsurface temperature and salinity, and most bio-parameters.

- Satellite profilers provide parameters which are either impossible to obtain from floats (global data on altimetry/sea level and waves, current, ocean colour, sea ice and winds) or complementing in-situ data.

- The parameters provided by satellites are key for ocean circulation models and assimilation techniques. Experiments have shown (see 3.2.2) that assimilating both satellite and in-situ data significantly improves sea surface temperature and salinity forecasts.

As explained by Roemmich et al. (2019), Argo data are used for education, research, monitoring and forecasting. The Argo programme must then be kept relevant to these different uses.

The main challenge facing Argo is now to sustain a long-term global coverage under budget constraints. New Argo development projects (Roemmich et al., 2019) include:

- The spatial expansion of Core Argo and regional increases in float density in poorly sampled zones;
- Achieving the long-term sustainability of BGC-Argo by increasing cost-effectiveness (decreasing sensor costs) and adapting the types of observations to users' requirements;
- The implementation of the Deep Argo programme, with a new design of the Argo array including -4,000 to -6,000 m observations.

## 2.1.2 Data acquisition, processing and supply chain

Argo is part of a supply chain stretching from primary data collection to processed data distribution and, further downstream, to value-added marine services for environment monitoring and security (Figure 5). Downstream from in-situ platforms, the data acquisition chain, after satellite re-transmission, includes a data assembly segment:

- Eleven national Data Acquisition Centres (DACs) collect, quality control, standardize, archive and distribute real-time (within 24 hours of data transmission from the float) and delayed-mode (within a year of collection) biological, chemical, physical and geophysical ocean profiles from Argo floats and other types of profilers, and convert data into standard exchange format.
- These data, together with associated metadata, are reported to the Global Telecommunication System (GTS) of the World Meteorological Organisation (WMO), to meteorological forecasting centres, and the two Global Data Assembly Centres (GDACs) in charge of collecting Argo data: FNMOC, USA, and Coriolis, France. Observational data are then made publicly available from GDACs via the internet.

Marine services are developed using a variety of processed marine data: research plays an essential part in such development. In Europe, Mercator Ocean International and the Copernicus Marine Environment Monitoring Service (CMEMS) play an essential role in this area. Mercator is an ocean monitoring service provider delivering customized real-time and delayed mode ocean information services on a periodical basis. CMEMS is an online service providing all users (service providers or end-users from the commercial and research and development sectors) with core information on maritime safety, coastal and marine environment, marine resources, and weather, seasonal forecasting and climate. It includes: 1) Thematic Assembly Centres (TACs) which process real-time in-situ and satellite observations; TACs include an in-situ component (INS TAC) which collects, quality controls and validates Argo and non-Argo data; 2) Monitoring and Forecasting Centres (MFCs) which run ocean models assimilating the data supplied by the TACs to generate analyses of the

current state of oceans, reanalyses (analyses of the past state of oceans using model and observations) and 10-day forecasts. At regional level, Argo Regional Centres (ARCs) provide expertise on specific ocean regions using Argo and non-Argo observations. The data generated by the Argo system are made freely available to the public within hours after collection. Standardized marine services include observation data compliant with the EU INSPIRE Directive 2007/2/EC in terms of data interoperability. For instance, EMODnet (European Marine Observation and Data Network), as a major component of the EU marine data supply strategy, publishes online interoperable data products and geographical information system (GIS) maps on bathymetry, geology, seabed habitats, chemistry, biology, physics and human activities (http://www.emodnet.eu). These marine data products are provided for research, archiving, observation and forecasting purposes to end-users, including operational end-users such as maritime transport (shipping, cruise lines) and offshore mining (oil rigs, dredging) but also commercial consultancies. The latter are active on customized service markets based on local forecast and specific coastal management services.

While the marine data acquisition and distribution market is mainly driven by public funding and investments, the US-originating strategy of free-of-charge supply of marine observation data products, aimed at supporting the competitiveness of downstream value-added services by decreasing their production costs, is superseding alternative business models outside the U.S., notably in Europe (see Groupe interministeriel, 1995). On the market of marine data products, the free-of-charge principle reinforces the role of public funding and public agencies in the supply chain, while business opportunities are created at downstream stages: in businesses such as "big data" management in Earth observation, and customized data products and value-added services, many private companies are now active (see: European Commission, 2017). CMEMS is part of the free-of-charge marine data supply chain. In 2016, the number of users of CMEMS data products was estimated at around 5,000, 80% of which were from the public sector. "Intermediate users", i.e. value-added service providers using and processing CMEMS data products, were large companies with a large market share but also small and medium-sized enterprises (SMEs), micro-companies and research institutions (PwC, 2016).

### 2.1.3 Summary on Argo

The above description highlights the horizontal and vertical integration of Argo with other components of the ocean data supply chain. Horizontally, the combined working of Argo, the other in-situ observing networks and satellite altimetry provides a full set of ocean observations. Vertically, primary observational data are validated and archived after quality control by the data assembly centres. An important part of the economic impacts discussed in this paper are generated by the combined working of Argo and of these components of the supply chain. Argo is therefore a major contributor to the benefits arising from ocean analysis, monitoring and forecasting, but works in association with other key components.

### 2.2 Cost of Argo

To assess the impacts of ENV RIs, we need information on their fixed and running costs for comparison. Regarding Argo, the AtlantOS "Optimizing and Enhancing the Integrated Atlantic Ocean Observing System" project (H2020, 2015-2019)

issued a report (Reilly et al., 2018) providing the estimated costs of a selection of ocean observing networks in the Atlantic, including the costs of the Atlantic Argo array (Euro- and US-Argo floats). Data gaps are recognised by the report as a limiting factor; pending a better data coverage, the costing exercise is thus a first step toward consistent accounts of Atlantic Ocean observing networks. However, as it stands, it provides valuable information on the costs of in-situ observing networks over the 2012-2016 observation period, with a focus on the chain from in-situ operations to data collection and transmission. Data management and downstream operations such as the development of data products are outside its scope.

The report provides cost estimates for the Argo floats of the AtlantOS area (Table 1), deployed by Euro-Argo ERIC members (Bulgaria, Finland, France, Germany, Greece, Ireland, Italy, Netherlands, Norway, Poland, Spain, UK), and by three other AtlantOS consortium members (Brazil, Canada and the US). The calculation methodology is based on the number of floats deployed per country. Table 1 shows only cross-country averages which do not reflect inter-country variations of unit costs. However, fixed costs show significant differences between the different types of float. The report also provides estimates for the 2017-2021 period in terms of expected investments and running costs. However, these estimates were tenuous at the date of publication, and have not been used in Table 1.

Regarding the 2012-2016 data, certain cost components would require further research:

- Staff costs have been excluded from the accounting and remain to be analysed. The challenge is for all participating countries to have a common definition of the activities involved in Argo operations. The report indicates differences between member states in this respect.
- The costs of research vessel services and those of ships of opportunity have also been excluded although data were collected for a few countries. Cost harmonization remains to be extended to this area too.

According to Table 1, the yearly costs of investments in floats are estimated at EUR 8.5m over the period of analysis. These estimates, limited to the Atlantic array of Argo, account for about a third of total Argo programme costs on the global scale, and for the same cost categories as in Table 1.

## 3 Approach to the assessment of economic impacts

Following the classification presented above, this section proposes an approach to the assessment of upstream, downstream and feedback impacts.

### 3.1 Assessment of upstream impacts

Upstream impacts are generated by the development and running of ENV RIs and on their requirements for equipment supply. This includes, inter alia, the construction of dedicated facilities and infrastructure (which may be very high); the manufacture and trade of remote sensing and in-situ measurement instruments, airborne or waterborne platforms; data transmission; data collection and processing; repair, refurbishment and other maintenance interventions. In this section,

attention is given to the assessment of upstream impacts in terms of business indicators such as turnover, value added and job creation.

### 3.1.1 Analysis of suppliers: direct assessment

In the favourable situation where suppliers are clearly identified and accept to provide economic information on their businesses, a direct assessment of upstream economic impacts is feasible.

A case study proposed by Barrow et al. (2005) on "marine science and technology industry" (MST) in New England (group of five US states) illustrates the approach. The paper analyses MST employment, sales and number of establishments per type of activity, for each state of New England.

The methodology of the case study uses a master database of 481 MST companies and organizations sourced from universities, public agencies and a commercial consultancy's database, and supplemented by web searches and targeted interviews. However, the companies listed in the master database are totally or partially involved in MST. To characterize this total or partial involvement, their employment and sales figures have been weighted at 100%, 40% or 10% if they pertain to the core, partial core or second tier respectively, of MST. This operation has led to a set of adjusted indicators (Table 2) for MST in New England. Companies are classified into five subsets analysed in terms of sales and employment. The "Marine Instrumentation and Equipment" (MIE) and "Marine Research and Education" (MRE) subsets are relevant to our approach; they include respectively: 1) cutting edge measuring instrument producers for oceanography, geophysics, acoustics, electronics for marine instruments, platforms and marine navigation; 2) marine education, industry and technology and transfer groups, and all areas of oceanography including fisheries research. The other subsets are less directly usable: "marine services" include consultancies and engineering; "marine materials and supplies" includes marine equipment (paints, engines, machinery, etc.); "shipbuilding and design" includes civil and defence shipbuilding operations.

Though its 2004 data would require updating, Barrow's approach points to a feasible way to assess the upstream segment of EU Ocean Observing RIs, based on targeted inquiries on jobs and sales of a delimited set of businesses. But Barrow's objective is to describe the industry of a given geographical area using a simple economic metrics, but not to assess the upstream impacts from ENV RIs on the supply industry which may be active beyond the geographical limits of New England.

A master database of MST companies such as the one used by Barrow would be worth developing for the EU. It would be a relevant start for further analysis on upstream impacts at an EU scale. An option would be then to define a set of activities equivalent to MST for EU countries and collect economic information through ad-hoc business inquiries.

Finally, note that Barrow's approach illustrates that ad-hoc surveys can help to analyse specific sector of the upstream supply segment; the survey is ocean-oriented, but the methodology is not ocean-specific.

### 3.1.2 Analysis of suppliers: indirect assessment

In less favourable but more frequent situations where suppliers are identified but do not provide commercial information, the impacts of ENV RI's operational and investment costs can be estimated using the standard classification of activities serving as a basis for national statistics. The EU statistical framework is based on the NACE[1], from which each member state derives its own national classification. In that case, the best option is not to rely on the official data as sourced from businesses:

- Enterprises are classified into NACE classes: one class per enterprise, based on its core business.
- "Structural Business Statistics" (SBS) are yearly updated by the National Statistical Institutes (NSIs) of the EU member states, using annual business inquiries. They provide financial and non-financial indicators (turnover, purchases, gross value added, number of employees, personnel compensation, etc.) per NACE subdivision. These primary data are processed to elaborate National Accounts, where economic consistency is ensured.

SBS are readily available to the public from Eurostat, the statistical division of the European Commission, which collects the data from NSIs and provides series of annual economic indicators per NACE class and member state (see https://ec.europa.eu/eurostat/data/database). If the classes including the targeted businesses (ENV RI suppliers) can be identified, it is feasible to estimate the impacts from ENV RIs' operational and investment costs in terms of turnover and employment, using SBS economic indicators as average estimates of suppliers' economic data.

There are two limits to this option. (1) SBS series are developed without time consistency objective. It is therefore preferable to use the branch accounts (of National Accounts) when they are available at class level: they allow for year-to-year comparisons. (2) Though class is the finest level of the NACE, many classes include a diversity of activities; in such cases the economic indicators of such classes do not permit to characterize a specific subset of businesses such as RI suppliers.

For instance, NACE class 26.51 "Manufacture of instruments and appliances for measuring, testing and navigation" includes the manufacture of aircraft engine instruments, radars, medical laboratory instruments, etc. The manufacture of ocean measuring instruments is only a modest part of this set. Likewise, the wholesale trade involved in the upstream segment of ocean RIs account for a small part of NACE class 46.69 "Wholesale of other machinery and equipment".

Contrary to ad-hoc surveys where the sample of respondents can be adapted, SBS and National Accounts break down economic activities into subsets (classes) that cannot be adjusted to the type inquiry. Consequently, only average estimates can be used to assess the upstream impacts from ENV RI development, using SBS and complementary indicators (such as yearly development costs, for instance).

---

[1] The "Classification of Economic Activities in the European Community" (referred to by its French acronym: NACE) is a hierarchical classification of EU economic activities into 21 sections, 88 divisions, 272 groups and 615 "classes". Member states' National Statistical Institutes develop national versions of the NACE adapted to their respective economies.

### 3.1.3 Analysis of ENV RI purchases

Where information on suppliers is missing, a consistent option is to collect the available data on ENV RIs' equipment and service purchases, either directly through ENV RI purchase records or indirectly through inventories and stock variations. This includes investments in infrastructure and equipment, as well as installation and maintenance services.

A standard statistical tool permits to classify the different types of products and services: the hierarchical "Classification of products by activity" (CPA), in force in the EU, classifies products (goods and services) by physical characteristics and by originating activity as defined in the NACE. CPA classes are strictly consistent with NACE classes: each CPA product corresponds to a specific NACE class. Like the NACE, each member state adapts the CPA to its national economy.

National accounts then permit to quantify the direct impacts of demand, as classified by products, on the corresponding

production branches. Such impacts are expressed in terms of incremental value added and workforce per NACE class, based on branch accounts. Indirect impacts, generated by suppliers' spending, raise other kinds of economic questions, typical of impact studies but not specifically linked with the upstream impacts analyzed in this section.

An example of ENV RI purchase data, limited to scientific equipment, is given by INSU (Oceanography division of the French National Centre for Scientific Research). INSU publishes an inventory of its open ocean instruments, available

online, including every type of material with technical specifications, purchase value, origin and number of instruments. The periodical flows of INSU purchases provide the data that would be necessary for assessing demand: flows can be inferred from stock variations based on successive inventory updates.

Table 3 summarizes INSU's inventory, limited to the amounts of purchase costs by originating country. It shows the diversity of INSU's suppliers, though more than half of the equipment value originates from North America and more than a

third from the US alone. A detailed assessment of the upstream impacts of European ocean RIs must therefore take account of imports, especially from North America.

However, note that (1) the example of INSU related to scientific equipment would not exclude examples related to purchases of other types of infrastructure components. (2) Such method is of limited accuracy if the branch accounts referred to above include large numbers of businesses: it then provides an average valuation of impacts. In any case, it is preferable to have the

branch accounts available at class level (the finest level of the NACE).

### 3.1.4 Summary of upstream impact assessment

The upstream impact assessment methods presented above are based on the combined use of data on ENV RIs' demand and supply business accounts. Each of the options examined above has its area of validity and its limits. Validity is determined by the relevance of sources and the quality of statistics. The limits are methodological: ad-hoc inquiries are time consuming

and require much input from respondents; SBS are useful proxies for suppliers' business data but must be used with caution as inter-annual comparison is impossible; in the absence of information on suppliers and if ENV RI purchases are detailed,

branch accounts are the most consistent tool to obtain an accurate valuation of upstream impacts in general terms, provided information is available at class level.

Practically, the selection of the most appropriate tools will depend on their capacity to analyse a diversity of supply business categories. Estimates are inevitable if business data are not available from enterprises; in such case, the resolving power of
the available statistics is critical for the quality of estimates.

## 3.2 Assessment of downstream impacts

This section considers the downstream stages of the environmental data supply chain, from primary data management (data assembly, quality control, archiving and assimilation for modelling, analysis and forecasting) to the data user categories, including public and commercial value-added services, and end uses.

The performance of the data from ENV RIs determines the quality of environmental observations supplied as primary material for the downstream chain. So, the analysis of ENV RIs' downstream economic impacts requires a performance assessment of the first stages of the chain: primary data collection, and processed data for monitoring and forecasting. In the absence of commercial transactions at these two first stages and given the free-of-charge principle (see 2.1.2), such performance assessment will be based on non-market indicators. Further downstream, most of the customized value-added
services perform commercial transactions: market related tools are necessary for performance assessment.

The subsections below focus on the two first stages mentioned above. Argo and the other components of the ocean observing system provide a practical example for the discussion. Additional remarks will then be made on value-added service markets.

## 3.2.1 Performance of observations: the role of key performance indicators

This subsection focuses on the case of Argo and other in-situ ocean observation arrays. But what is said below can be generalized to a range of ENV RIs collecting data from environmental observations.

At the stage of observational data acquisition, archiving and quality control, DACs and GDACs are key players. GDACs publish information on data performance using a set of key performance indicators (KPIs).

The KPIs used by Coriolis, other assembly centres and INS TAC (see 2.1.2) constitute a metrics of the quality of ocean
observations, also resorted to by CMEMS for performance monitoring; they include:

-   delay (e.g. share of delayed data per time interval),
-   Types and number of platforms (by type of sensors and by measured parameter),
-   Accuracy of measurement (for temperature and salinity),
-   Number of downloads and number of users.

Certain KPI series do not go back many years in time and cannot characterize long term progress in equipment generations. However, the number of platforms per type or per parameter is monitored since 2014 by Coriolis, which is enough to give indications on trends. Likewise, several other monthly, weekly or daily KPIs generate sizeable time series with short term

monitoring purposes, as shown by KPIs available from INS TAC at http://www.marineinsitu.eu/monitoring/. In the future, longer series will enable to analyse correlations between data supply and demand, i.e. between data quality KPIs and the number of downloads.

Complementary sources of KPIs are available, e.g. from the Joint WMO-IOC Technical Commission for Oceanography and

Marine Meteorology (JCOMM), an intergovernmental body in charge of, inter alia, co-ordinating the WMO Marine Meteorology and Oceanography Programmes. JCOMM's "in-situ Observation Programme Support Centre" (JCOMMOPS) monitors, and provides metrics for, a range of in-situ observing networks including Argo. Indicators on the array (activity, density, intensity), data flows (delivery, sensor metadata quality, timeliness) and national diversity are made available to the public on the JCOMMOPS website (https://www.jcommops.org/board). Some data series started in 2000, which secures a

correct historical background. Observing float arrays monitored by Coriolis and JCOMMOPS overlap; some measurement differences may arise between the two sets of floats.

Beyond the specific case of Argo, such assessment method is based on standard KPIs and could be generalized to other ENV RIs: as pointed out by Roemmich et al. (2019), the Argo data management system has been used as a model for other observing systems.

**3.2.2 Performance of analysis and forecasts: the role of observing system experiments**

This subsection focuses on issues relating to ocean and atmosphere RIs. It can be generalized, with caution, to ENV RIs generating primary observational data for processing and environment monitoring and forecasting.

At the stage of environment modelling and forecasting, ocean RIs rely on real time observations. To assess forecasting accuracy, "observing system experiments" (OSEs) have become a frequent tool. Originally designed and performed for

atmospheric observations, OSEs were adapted to ocean observations. Their interest lies in that the use of observations is critical for ocean modelling and forecasting; the performance of an observing system must then be assessed based on its impacts on ocean models and forecasts.

The traditional OSE method consists in running an ocean model under various observation use scenarios: a given subset of past observations (e.g. satellite altimetry or part of the Argo profiles) is withheld from an assimilating system, for a given

ocean model. The resulting forecast error on real observations allows indirectly assessing the impact of assimilating the subset of information withheld, for a given assimilation method. Based on a similar methodology, "observing system simulation experiments" (OSSEs) are used to quantify the potential impacts of new assimilation techniques and new generations of observation instruments. OSSEs use simulated observations from a free-running model, called "nature run". Used as environmental truth, these simulations are assimilated in the ocean model; system experiments are then performed

by comparing the forecasts generated by the model with simulations from the nature run.

In the recent past several OSE-based studies analysed the impacts of Argo on ocean observing systems and forecasting (Turpin et al. 2016). In the framework of a research project on the impacts of Euro-Argo, a range of contributions using OSEs and OSSEs were summarized in Rémy and Le Traon (2015). OSEs were also applied to GOOS components, including

satellite altimetry, Argo and non-Argo floats (Oke et al., 2015a), and to regional observing systems, in order to provide the GOOS with additional constraints for initialising ocean models, and to resolve regional and local processes (Oke et al., 2015b).

The traditional OSEs presented in these publications calculate the root mean square of the difference (RMSD) between forecast and real-time observations to measure the impacts of a specific set of observations. Turpin et al. (2016) assesses the impacts of Argo data assimilation on the short-term real-time analysis and forecast of sea surface temperature and salinity. It summarizes OSEs conducted over 2012. These OSEs include the assimilation of observations from satellites, from all other-than-Argo in-situ instruments and from a share of the Argo array (100%, 50% or 0%). The results show that the impacts of Argo profiles on RMSD for temperature and salinity are significant from sea surface to -2000 m: the use of Argo data can lead to a 20 to 50% decrease in RMSD for temperature on the 700-2000 m depth layer, and 30 to 65% for salinity.

More generally, the findings from Turpin et al. (2016), Oke et al. (2015a and 2015b) and Rémy-Le Traon (2015) emphasize the significant impacts of Argo. They include the following points:

- The diversity of OSEs, though based on different approaches, permits to conclude that the use of Argo profiles has positive impacts on real time ocean analysis and forecasting in a wide range of cases including short- and long-term analyses, ocean properties up to 2000 m depth, and sea level analysis.

- Regarding sea level analysis, Argo observations on sea water properties are critical to complement satellite altimetry and constrain the ocean state analysis.

- These remarks must be qualified: the results may vary depending on sea water parameters and the state of deployment of Argo. For certain parameters and certain ocean regions, non-Argo observation networks are as important as Argo.

- The results obtained from OSEs and other experiments depend on the ocean models and data assimilation systems which are used, as emphasized by Turpin et al. (2016) and Oke et al. (2015b). It is therefore impossible to draw general conclusions in quantitative terms, from each specific experiment.

- Monthly OSEs were also carried out in excluding different specific subsets of observations, e.g. XBTs, Jason altimetry, Tropical Pacific buoys, Argo, etc. (Oke et al., 2015a). Results evidence the complementarity between the different subsets in terms of forecasting performance: this important conclusion goes beyond an Argo vs non-Argo comparison.

Observing system experiments are therefore an important tool for assessing the impacts of the GOOS in terms of forecasting performance. The method is also relevant to analyse the expected impacts from the use of new generations of instruments. It must also be stressed that the quality of models and assimilation techniques – i.e. outputs from research – is key for forecasting performance: OSEs appear as a performance indicator for the combined contribution of several segments of the supply chain.

### 3.2.3 From ENV RIs' performance to downstream impacts

The previous subsections focused on the first stages of the downstream data supply chain. This subsection is an attempt to consider the methods used to address the impacts on end-users, including commercial value-added services using environment monitoring data. Argo, as an ocean RI, will serve as an example but the focus is on ENV RIs.

Assessing ENV RIs' impacts on the value-added service segment of the supply chain requires business and market indicators, e.g. the amount of demand for primary and processed data. Additionally, it would require studying correlations between performance as described by the KPIs discussed above and the size of downstream markets if information was available. This subsection briefly looks at these issues.

- As said above, the KPI series developed by the DACs (subsection 3.2.1) include indicators such as "number of users" and "number of downloads" which characterize the amount of demand. Such indicators should be supplemented by detailed information on users (types of company or institution, type of use, etc.), in order to further analyze demand trends: similar information on data users was collected by PwC (2016) on the Earth Observation downstream market.

- Further downstream, the PwC (2016) report offers an example of processed data market analysis. The work is based on an inquiry on a substantial sample of CMEMS data users and their business characteristics. It shows the feasibility of assessing the downstream impacts on processed data product markets: the inquiry helps to classify the different types of data users and evaluate the size of the ocean-related value-added service market, based on income generated to service providers. For instance, the 2015 income was estimated at EUR 5.6 to 8.6m. It is expected to grow substantially and reach nearly EUR 28m in 2020. Similar inquiries, performed on a periodical basis, would allow generating a relevant database complementing information on downstream impacts.

These two sets of indicators are adapted to the features of the supply chain: the free-of-charge principle prevailing for the first stage of ocean data processing makes it impossible to refer to business statistics based on turnover or benefits; digital information resources offer an alternative. The indicators are also adapted to the lack of official business statistics on value-added service providers: targeted inquiries are relevant alternatives.

The question as to how to assess linkages or correlations between the first stage performance indicators and the downstream market indicators remains to be resolved. At this stage, to our knowledge, the lack of data does makes it difficult to study such correlations. The question must be left open at this stage.

### 3.2.4 Downstream impacts on knowledge development

As said earlier, OECD (2016) proposes a list of indicators which can help to quantify the general impacts of RIs. OECD's assessment consists of a list of 25 "core impact indicators" and 33 "additional indicators", most of which are quantitative, and classified by RI objectives (e.g. promote education outreach and knowledge transfer; provide scientific support to public policies) and by main impact categories (e.g. scientific impact; technological impact; training and education). Indicators

include, inter alia: number of citations received by publications using the RI; number of projects funded by external grants; number of projects developed with other RIs; number of PhD students from universities using the RI; and number of staff in full time equivalent.

Such indicators are relevant for an indirect assessment of the long term economic impacts of Argo. The Argo community is concerned to keep the program relevant for the diversity of users to "deliver societal benefits" (Roemmich et al., 2019). It points out research and education as key users, and resort to the metrics mentioned above to appraise their significance. Riser et al. (2016) uses the number of publications since the beginning of the Argo program (2,100 papers in the refereed science literature until 2016); Roemmich et al. (2019) uses the number of Ph.D. having used Argo data (about 300).

This illustrates the dual-use nature of an ENV RI such as Argo, with operational uses in terms of ocean monitoring and forecasting, and scientific and educational uses with long term objectives.

### 3.2.5 Summary of downstream impact assessment

The above summarizes practical approaches to assessing the performance of ocean data acquisition and of ocean analysis and forecasts.

- KPIs, as applied to in-situ observations, provide combined information on observational data quality, the number of platforms and the size of demand (number of downloads). Generally, the set of relevant KPIs will depend on the nature and objectives of ENV RIs.
- OSEs are considered herein in relation to ocean forecasts. They prove useful for assessing a) forecasts, based on the combined use of in-situ and satellite data; b) the contribution of Argo to the performance of forecasts. In the future, it will be essential to have OSEs analysing the contribution of the new components of ocean observing systems, e.g. Deep Argo.

KPIs and OSEs have different scopes of application. Combined together, they are a valuable step toward the assessment of ocean RIs' downstream impacts. Periodical performance monitoring based on routine OSEs would be an essential complement for assessing the value of the GOOS at its present and future stages (cf. Oke et al., 2015a; Rémy and Le Traon, 2015; GOV Science Team, 2014).

Further downstream, more data will be necessary to study the impacts on ocean data users, including value-added service providers, and analyse correlations between performance indicators and demand for ocean data products.

Finally, it is important to monitor the use of ENV RIs by research and education, which gives indications on their long term economic impacts and societal benefits. Indicators are available in this purpose, some of which have been used to appraise the importance of the Argo program.

### 3.3 Assessment of feedback impacts

This section focuses on the impacts of the entire data supply chain of an ENV RI on environmentally sensitive users. The additional knowledge provided by observation and processed data is used by businesses and other stakeholders for increasing

efficiency, welfare and competitiveness. Such information can also be used to better respond environmental hazards and uncertainty. This section will focus on the benefits to users from environmental monitoring, data acquisition, and forecasting. Cost-benefit analysis (CBA) is a classic tool to test whether an ENV RI project serves public interest. It consists in calculating the benefits to information users – in terms of increased welfare, efficiency and avoided costs – minus the costs of the project. This section focuses on methodological issues raised by the assessment of feedback impacts from ENV RIs and discusses examples of CBAs on weather and ocean observing systems. It addresses some specific issues raised by the case of Argo, but its purpose is not to make a CBA on Argo in particular.

### 3.3.1 Examples of CBAs on ocean observing systems

Since the 1990s, several CBAs were performed to assess the expected impacts of ocean observing systems. Four examples of CBAs are summarized below. The three first ones were published at an early stage of the development of in-situ ocean observing systems and were intended to evaluate the relevance of public support to the investment projects. A fourth example, published later, was related to the future GMES Global Monitoring for Environment and Security (renamed Copernicus in 2013), with the same evaluation purpose.

- Sassone and Weiher (1997) focus on the El Niño Southern Oscillation (ENSO) phenomenon and its impacts on the U.S. agricultural sector and analyse the potential impacts of improved ENSO forecasts.
- Solow et al. (1998) also analyse the value of improved ENSO forecast to U.S. agriculture, but with a focus on cropping strategies.
- Kite-Powell and Colgan (2001) consider the local marine activities in the Gulf of Maine which could use and benefit from the development of an ocean observing system: maritime transport, commercial fishing, recreational fishing and boating, SAR, pollution and oil spill management.
- Cedre[2] and Ifremer (2009) analyse the potential impact of GMES monitoring and forecasting services on the prevention and mitigation of a past oil spill event

These CBAs rely on methodological options concerning: a) the categories of data users to consider in the analysis; b) the definition of the notion of "improved forecasts" with reference to a baseline scenario which must also be defined; c) users' sensitiveness to observations and forecasts, and their adaptation capabilities, in terms of consumption or business strategy, to new sets of environmental information; d) the spatial scale and time period of the analysis.

Sassone and Weiher (1997) build on progress in ENSO forecasting capabilities and on the findings of the Tropical Ocean Global Atmosphere (TOGA) project (1984-1995) on coupled tropical ocean-atmosphere modelling. The aim of the paper is to discuss the relevance of a projected, government funded, research program combining an extension of TOGA and an ENSO Observing System (EOS) to make routine ENSO forecast. To do this, a CBA assesses the development of TOGA-

---

[2] Cedre: French state agency created in 1978 after the oil spill from Amoco-Cadiz oil tanker. It provides technical advice and expertise to French and foreign authorities and businesses in charge of responses to marine accidental pollution, particularly oil spills.

EOS versus a baseline scenario in which the research program is not carried out. Projected costs are based on past TOGA costs and government's expectations for EOS costs; projected benefits are those to the U.S. Agriculture sector. The overall benefits to agriculture are the sum of farmers' profit and consumers' surplus, farmers being assumed to gradually accept forecasts and adapt their strategy accordingly: gradual adaptation is translated in quantitative terms. Annual costs and

benefits to the agriculture sector are estimated for 10-year and 20-year periods from 1995. This allows calculating an internal rate of return (IRR)[3].

Solow et al. (1998) focuses on farmers' cropping strategies: the yields of the different types of crops depend on climate conditions which in turn depend on ENSO phases (i.e. warm or cold climatic phase, or no climatic event). Farmers therefore optimize benefits by adjusting their cropping strategies (or crop mix) to the forecasts of ENSO phases per U.S. agricultural

region. Using a meteorological model and a plant growth model, the method calculates the economic surplus from an ENSO phase prediction (i.e. benefits minus losses to producers and consumers) in a scenario with such prediction versus a baseline scenario without prediction.

An essential feature of Solow's methodology is that a statistical model is used to take prediction skills into account, as expressed by a probability of realization of predicted phases. Farmers adjust cropping strategies by maximizing their

expected surplus, given a range of ENSO phase predictions. An economic model of the agriculture sector is used to calculate the expected surplus resulting from changes in crop yields and crop product markets (prices and quantities). The economic surplus is then summed over all phase predictions weighted by the probability of realization of each prediction.

Kite-Powell and Colgan (2001) focus on some of the major maritime activities in the Gulf of Maine: maritime shipping, commercial fishing and fish farming, boating and leisure fishing, search and rescue, and oil spill management. Ocean

observations are generated by the observing system of the Gulf of Maine (GoMOOS). The study differs from the two preceding examples by its focus on the general impacts of a regional ocean observing system on a set of maritime activities. The potential benefits of additional ocean information are expressed in terms of lower operating costs and higher value-added for businesses, lower travel costs for recreational activities, and lower management costs of accidental pollutions due to more efficient damage prevention. For certain of these activities, potential benefits are difficult to quantify as the type of

usable data that the observing system would produce still need to be defined.

In a report for the InterRisk research project funded by the European Commission, Cedre and Ifremer (2009) focuses on the impacts of improved information on coastal seas on the response to a specific oil spill incident. The aim of InterRisk was to develop a pilot project of GMES (Global Monitoring for Environment and Security) monitoring and forecasting services for environmental risks and crisis management in marine and coastal areas. The study benefitted from the experience gained by

Cedre in oil spill management: Cedre has acquired a significant amount of information and data on damage and response costs generated by oil spill incidents, and on the impacted activities; it intervened during the wreckage of the Erika oil tanker

---

[3] IRR: discount rate at which a project, given its yearly costs and benefits, has a net present value of cumulated cash flows of zero at the end of a given time period. If the real discount rate is expected to be less than IRR over a period to come, real benefits are expected to exceed real costs at the end of the period.

in the Bay of Biscaye (December 1999): a 40,000 tonnes oil spill impacted a large part of the Bay of Biscaye coast, including natural and tourist sites, ports, marinas and fish and fish farming zones. The Erika incident was used as a case study: 1) Costs and losses incurred by marine and coastal activities from the Erika incident are estimated, based on claims and administrative data. The activities included in the assessment are fisheries, aquaculture, salt production, coastal hotels and tourist visits; impacts also concern coastal private and natural assets, and equipment and material supply to fisheries and aquaculture, seafood processing and local tourism. Public response to oil spill is also part of the costs. 2) A potential mitigation of the spill was estimated, expressed in percentage of the oil spill volume, as a result of GMES ocean forecasts being available to the state response scheme. Oil spill managers are thus supposed to be the only ocean data users. 3) Avoided losses for coastal activities, expressed in percentage of the total losses incurred in the baseline scenario (no GMES), as a result of damage mitigation.

### 3.3.2 Summary on CBAs on ocean observing systems

The CBAs examined above are ex-ante analyses: they assess the potential economic impacts in scenarios where more information is available to users. This requires assumptions on the definition and potential use of the information generated (e.g. what type of forecasts or improved forecasts would be made available to which users?). This also requires assumptions on how data are used by potential users. These two categories of assumption help to classify the above CBAs into two main categories.

The first type of CBAs (Sassone and Weiher, 1997; Solow et al., 1998), considers a single activity at country scale, namely the agriculture sector. Its development strongly depends on climatic conditions, including ENSO phases. In terms of methodology, a systematic analysis of the value of predictions for cropping strategies can be made within the framework of an economic model of decision making under uncertainty, using long data series. Better ENSO predictions lead to more efficient decisions and to an increase in production. After market clearing conditions, this leads to an increase in the expected economic surplus to farmers. The analysis can be extended to formalizing farmers' gradual acceptance of predictions over a long enough period. Gradual acceptance is expressed by an increasing share of the population of famers using the forecasts. This determines the evolution of the overall economic surplus over time.

The second type of CBAs (Kite-Powel and Colgan, 2001; Cedre-Ifremer, 2009) considers several categories of ENV RI data users, the spatial scale being limited to a given region. The analysis relies on a diversity of expertise, as several categories of maritime activities are included in the assessment. In the studies of the second type, there is no specific assumption on users' strategies and their adaptation to forecasts. In the Cedre-Ifremer study, the oil spill management authority is the only data user, and immediately adapts its strategy to the set of information provided by the new ocean observing system, in a situation where the time frame for oil spill response is necessarily tight.

In short, one method focuses on the impacts on a single category of users. These users improve their business strategy and mitigate environmental risks, using a set of improved but imperfect observations and predictions: users' responsiveness is key for explaining the benefits (avoided costs) generated by predictions. The other method considers several user categories,

supposed to instantaneously adapt to the new set of environmental data. Given the diversity of users, A wide range of expertise is required to estimate benefits per category.

### 3.3.3 The question of the feedback impacts of Argo

Such classification is useful to approach the case of Argo, considered as a separate entity, and to examine the methodological issues linked to the assessment of its feedback impacts.

While the above CBAs analyse the impacts of an entire ocean observing system, an assessment limited to Argo would focus on the specific impacts of a part of it: the assessment methodology necessarily differs from what was discussed above, but can be designed using the four examples of CBA. We suggest a three step approach: 1) measuring the impacts of Argo data on ocean models and forecasts in quantitative terms; 2) analysing the impacts of the improvement of ocean forecasting on ocean data users' economic behaviour; 3) analysing the resulting changes in terms of increased efficiency, risk avoidance and overall costs and benefits.

- First step: a relevant option is to refer to the method outlined in section 3.2.2 on downstream impacts. Appropriate observing system experiments can serve to quantify the impacts of Argo in terms of forecasting accuracy, measured by the mean error of forecasts. The experiments considered in 3.2.2 permit to conclude that Argo data significantly improve ocean circulation models and forecasts, but these experiments were carried out within short time periods (within a given year or even a given month). They depend on ocean models and data assimilation systems, and the quantitative results cannot be generalised. Drawing general conclusions from the available studies is impossible at this stage, and requires, as suggested by Oke et al. (2016a) and Turpin et al. (2016), obtaining more quantitative results from future experiments.

- Second step: if the above difficulty is solved in the future, a major question emerges as to how to identify the diversity of data users and assess their sensitiveness to the improved accuracy of forecasts. The selected CBAs address the impacts of ocean observing systems on specific data users (agriculture, oil spill managers, and marine activities). But a CBA on Argo would involve a diversity of user categories and require a baseline scenario describing users' behaviour under a non-Argo ocean observing system. But, to our knowledge, information is missing on users' sensitiveness to forecast accuracy improvement, pending long-term behavioural observations on marine activities. In terms of methodology, it remains to examine whether the statistical model used by Solow et al. (1998) can be adapted to several categories of users with quite different behaviour towards forecasts. Alternatively, Kite-Powell and Colgan (2001) considers several groups of marine businesses sensitive to improved ocean information, but the analysis is limited to the impacts of two opposite scenarios: with vs without an ocean observing system. The problem would then be to assess, using such approach, how sensitive marine businesses would be to certain components of the system.

- The third step requires estimating how users' behaviour (their willingness to use new datasets) will affect their benefits: e.g. prompt willingness to use improved forecast would better contribute to risk mitigation. Such assessment would require many economic data and a solid expertise on users.

Beyond the uncertain feasibility of a CBA on Argo, these methodological remarks give an overview of the material needed for the exercise.

## 4 Conclusion

The option taken by this paper is to subdivide the economic impacts of ENV RIs into upstream, downstream and feedback impacts. The paper provides an overview of the main assessment methods currently used for each impact category. Argo is an example of ocean RI which helps to address a wide spectrum of methodological issues, due to its specificities in terms of
equipment and infrastructure, and the diversity of data users and processors.

The assessment of upstream economic impacts is an essential step as growing environment concerns require adapting infrastructure and equipment technology in order to improve the research and monitoring capacities offered by ENV RIs. The assessment methods require detailed economic data both on upstream supply businesses and ENV RIs' demand. However, data availability is one of the main practical difficulties, especially in cases where information is limited by
commercial confidentiality; estimates are inevitable.

The methods of downstream impact assessment depend on the different uses of the ENV RI data products, i.e. they concern the different stages of the data supply chain, including observational data collection, environmental monitoring services and customized value-added services. Environmental risks and climate change concerns are important drivers of users' and stakeholders' rising demands for such products and services. The assessment methods examined above are mainly based on
non-monetary performance indicators. A key issue of this category of assessment is the need for analyzing correlations between (1) such indicators, (2) the demand for ENV RI data and (3) the size of the value-added service segment. In other terms, performance indicators are useful to analyze data quality in details, but there is a need for broadening the scope of the approach and assessing the impacts of ENV RI performance on the overall data and service supply chain.

The examples of feedback impact assessment examined above are based on CBAs in order to compare the benefits to users
with ENV RI costs. Improved weather and ocean forecasts as enabled by certain ENV RIs and downstream monitoring and forecasting services provide more knowledge on environment usable by several categories of stakeholders. Risk avoidance is becoming a major objective for environmentally sensitive activities and gives relevance to such CBAs. The discussion of feedback impact assessment methods is likely to focus on ENV RIs' incremental development, as shown by the current and future expansion of Argo. This raises again the question as to how to assess the impacts from developing a subset of an
observing system.

The above illustrates that the methodological issues remain numerous. The recommendations mentioned above concern (1) the need for more detailed data on upstream supply businesses and a monitoring of ENV RIs' purchases; (2) the need for

analyzing correlations between performance indicators and demand for data; and (3) the need for a better knowledge of environmentally sensitive user categories and their responsiveness to the availability of improved environmental data.

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

```
┌─────────────────────────────────────────────────────────┐
│                  Equipment and services                 │
└─────────────────────────────────────────────────────────┘

┌─────────────────────────────────────────────────────────┐
│                      Infrastructure                     │
│                     Research vessels                    │
│         Data management facilities and equipment        │
│        Data transmission facilities and equipment       │
│       Sensor-equipped platforms, floats and gliders     │
│                  Animal born instruments                │
│                 Satellite born equipment                │
│          Deployment and maintenance, ship services      │
└─────────────────────────────────────────────────────────┘
```

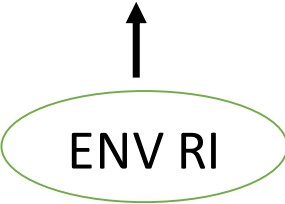

**Figure 1. Upstream impacts: list of the main categories of equipment and services supplied to ENV RIs**

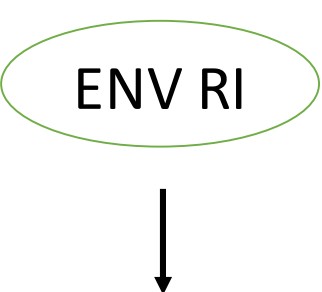

Observational data assembly, control and management
Data processing for analysis and environment modelling
Environment monitoring services and forecasting
Customized services, consultancy services

**Figure 2. Downstream impacts: the different stage of data and service supply from observational and processed data to the different environment monitoring services**

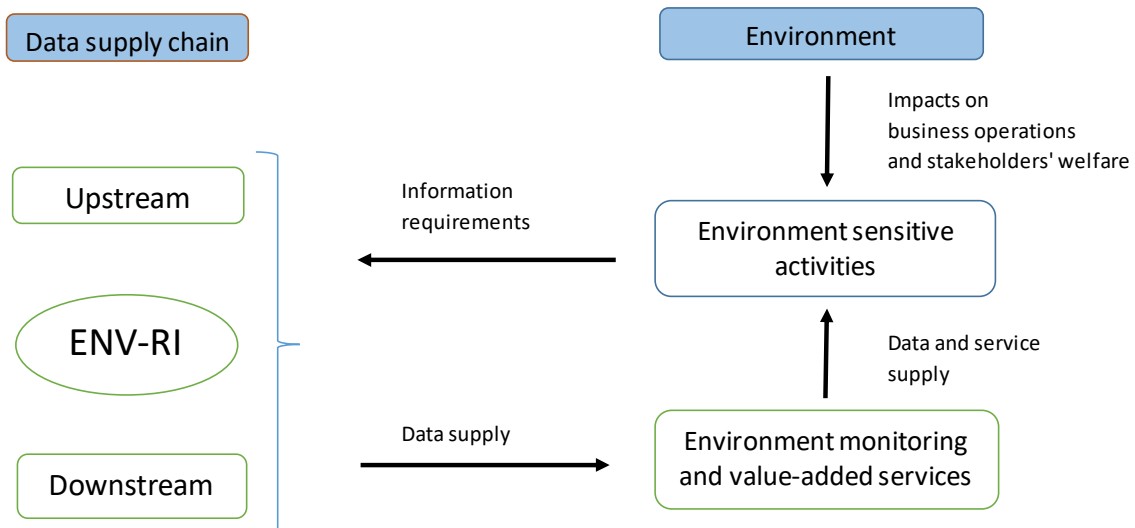

**Figure 3. Feedback impacts: environmentally sensitive activities require environmental information and are supplied with data, monitoring services and forecasts by e.g. Copernicus and other value-added services.**

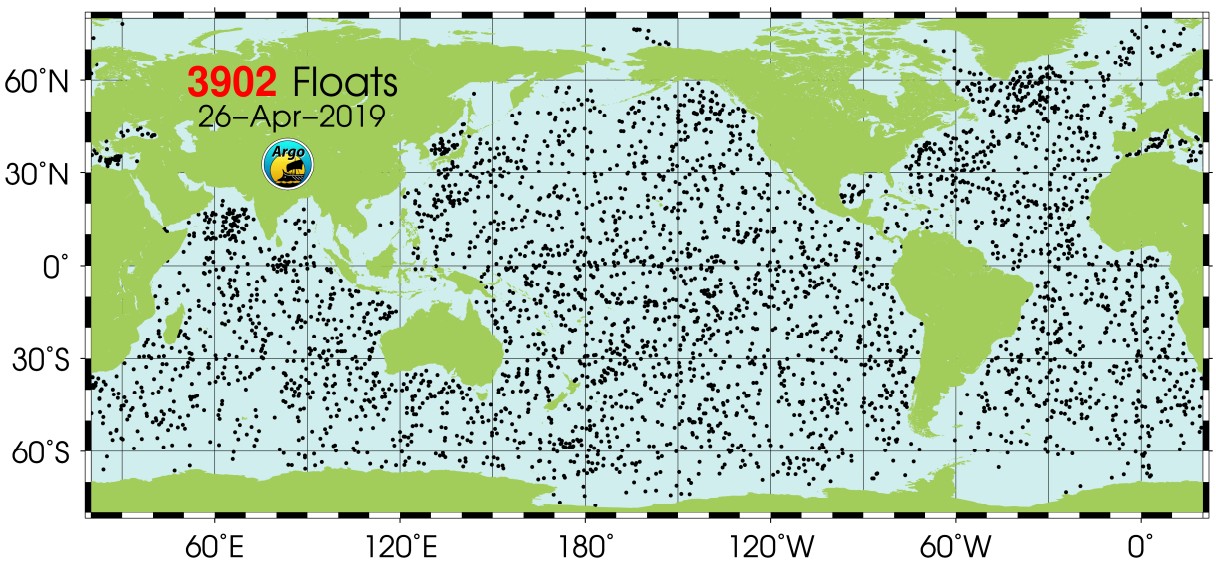

**Figure 4: Argo global array of profiling floats: position of floats having delivered data within the last 30 days until 26 April 2019.**
**Source: © Argo Program http://www.argo.ucsd.edu**

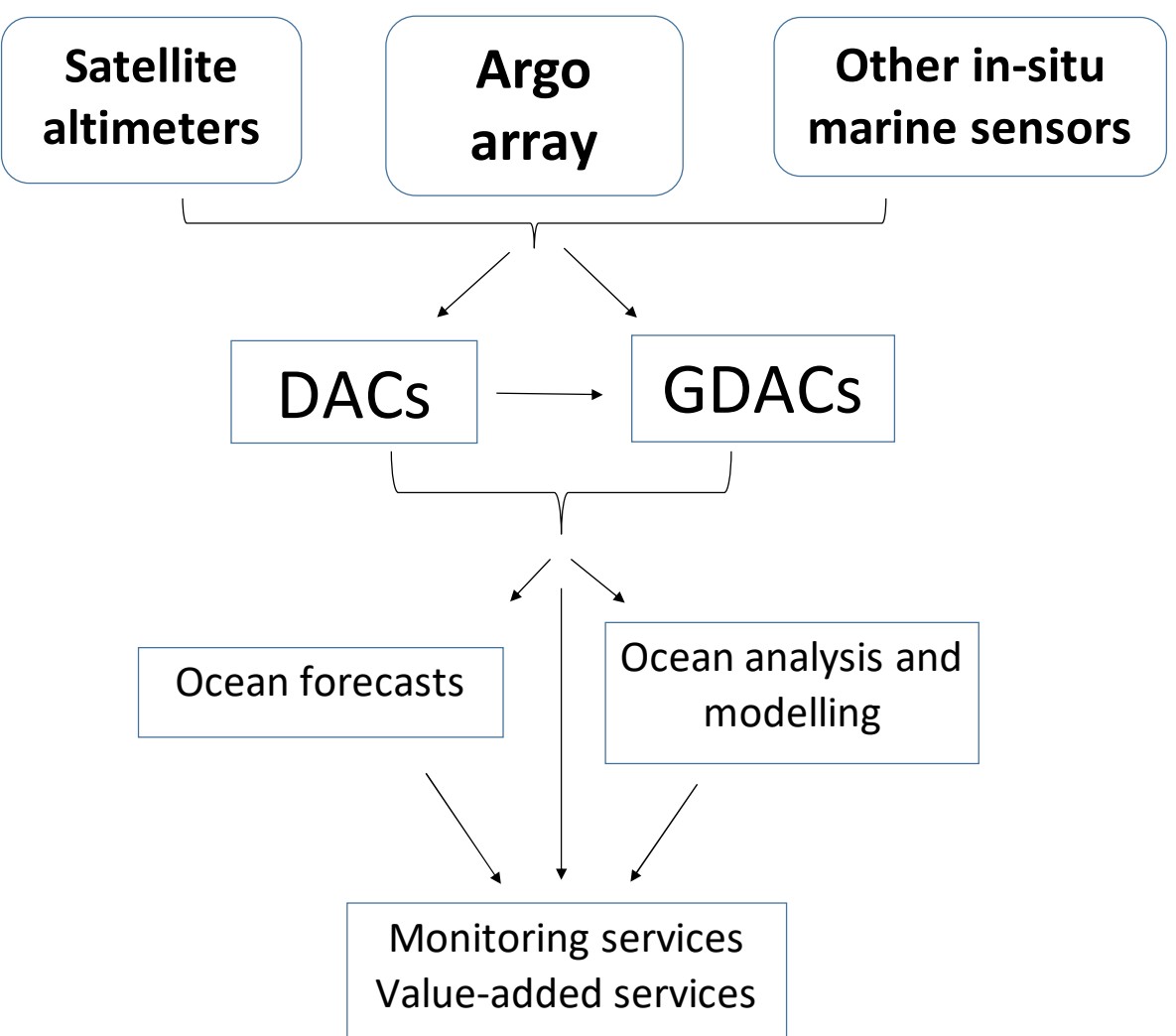

**Figure 5. Simplified description of the ocean data supply chain including the Argo array: observational data are provided to the DACs and GDACs, then to scientific users for processing and modelling. Observation and processed data are used for monitoring services.**
**DAC: Data Assembly Centre.**
**GDAC: Global Data Assembly Centre.**

| Estimated annual costs Time period of analysis: 2012-2016 | Core Argo (T&S) | BGC-Argo | Deep Argo* |
|---|---|---|---|
| Average yearly number of floats deployed** | 229 | 38 | 35 |
| Average unit cost (€) | 15 890 | 85 000 | 30 000 |
| Additional unit costs/year | | | |
| Testing and calibration (€) | 300 | 300 | 300 |
| Logistics (€) | 400 | 600 | 400 |
| Transmission (€) | 1 440 | 1 440 | 1 920 |
| Total (€) | 4 128 870 | 3 337 160 | 1 124 900 |

**Table 1: Argo programme costs for the Atlantic Ocean observing network**
**T&S: temperature and salinity profiler**
**BGC: biogeochemical profiler**
**\*Arvor Deep Argo floats only. Apex floats excluded.**
**\*\*Target annual deployment for Deep Argo and BGC floats.**
**Source: Reilly et al. (2018).**

| | MIE | MRE | MST |
|---|---|---|---|
| Employment* | 12496 | 1457 | 38906 |
| Sales ($ million) | 1966,3 | 6,8 | 4868,9 |
| Establishments | 175 | : | 481 |

**Table 2: Marine Science and Technology industry adjusted figures for New England, 2004.**
**: not reported**
**\*number of persons employed**
**MIE: Marine Instrumentation and Equipment**
**MRE: Marine Research and Education**
**MST: Marine Science and Technology industry**
**Source: Barrow et al. (2005)**

| Originating country of supply (1) | Value of material (2) ('000 EUR) | Market shares |
|---|---|---|
| US | 1416.7 | 37.0% |
| UK | 115.6 | 3.0% |
| Canada | 599.3 | 15.7% |
| Norway | 449.2 | 11.7% |
| France | 917.4 | 24.0% |
| Germany | 61.5 | 1.6% |
| Italy | 18.5 | 0.5% |
| Not specified (3) | 246.1 | 6.4% |
| TOTAL | 3824.3 | 100.0% |

**Table 3: Open ocean equipment, INSU Brest Centre**
**(1) Country of supplier's head office. Does not indicate country of manufacture.**
5    **(2) Purchase value for INSU.**
**(3) Material listed without details about suppliers.**
**Source: INSU.**

| Study | Topic | Methodology | Results |
|---|---|---|---|
| Sassone, Weiher, 1997 | Costs and benefits from TOGA project and ENSO observing system (EOS) | Impacts on US agriculture over 1995-2015. Sensitivity analysis using: -Producers' and consumers' skill level (capacity for adapting to forecast), -Rate of acceptance of ENSO forecast, -Annual future costs of EOS. | IRR = 13 to 26% Depends on assumptions on sensitivity of parameters. |
| Solow et al., 1998 | Benefits from better ENSO forecast on US agriculture through more efficient cropping. | Based on simulation: -Meteorology model for simulating ENSO forecast on temperature and precipitations. -Plant growth model for optimization of crops. -Economic model for impacts from crop strategies on crop product markets. Period of analysis not specified; annual expected impacts are given; an estimate is made over 10 years. | Expected producers' and consumers' annual surplus: $240 to 323 m as compared to ENSO forecast costs ~ $12.3m/year. |
| Kite-Powell, Colgan, 2001 | Benefits from GoMOOS on marine activities in the Gulf of Maine | Review of Gulf of Maine commercial and non-commercial uses of GoM waters: -Key indicators per activity: operating costs/day, value added/day, willingness to pay for leisure, oil spill cost reduction. -Period of analysis not specified. -Yearly estimated avoided costs per activity result from efficient use of ocean forecasts and improved business management. | Annual potential benefits = sum of avoided costs per activity. Sum estimated at $33 m: lower bound, owing to missing data for several terms. |
| Cedre, Ifremer, 2009 | Benefits from pilot tool for GMES. Case study: oil spill on France's Atlantic coast, December 1999 | Review of local marine activities impacted by the oil spill (commercial activities only). -Estimates of turnover and employment per commercial activity. -Estimates of avoided costs from more efficient mitigation of damages. -Estimates based on experience gained by Cedre on series of oil spills. -Period of analysis: approximately one year. | Sum of avoided costs per activity. Avoided costs: €49 m (conservative estimate) as compared to total oil spill cost estimate: €450 m. |

**Table 4. Four examples of cost-benefit analyses of observing infrastructures used for ocean forecasting and climate monitoring**

5 **TOGA: Tropical Ocean Global Atmosphere**
**ENSO: El Niño Southern Oscillation**
**GoM: Gulf of Main**
**GoMOOS: Gulf of Maine Ocean Observing System**
**GMES: Global Monitoring for Environment and Security**

**Annex**

Author contribution: the author drafted the whole paper as submitted.

Competing interests: the author declares that he has no conflict of interest.

Acknowledgement: the author is grateful to the ENVRI PLUS project for making it possible to carry out the work summarized in the present article. He is grateful for the assistance of the referees and the Euro-Argo team, including… in the preparation of this article.

