# Peer review of "Assessing economic impacts of environmental research infrastructures: overview of methodological tools"

_Geoscience Communication, 2019_

## Referee Comment (RC1) · Dwight Owens (Referee) · 30 Jul 2019

General Comments ——————————

I carefully reviewed your paper together with my colleague Kim Juniper, Chief Scientist, Ocean Networks Canada. These comments reflect our combined reactions.

We found your paper to be thought provoking and topical. The issue of assessing economic impacts of ENV RIs is important, as many organizations struggle to justify the investments required to establish and maintain their facilities. Improved methodologies for estimating economic impact will be of great interest to the operators of ocean ob-

serving facilities in many countries, since public money is often critical for continued operations.

A primary issue that stood out was the treatment of ENV RIs as purely operational facilities. Many (all?) big science infrastructures address both basic science and applied applications. For example, a large part of what our facility (Ocean Networks Canada) does is in support of basic science; there may be some eventual spillover into forecasting or hazard mitigation, but the emphasis is on supporting the basic research community, not the operational community. Also, many large infrastructures support educational audiences and outreach activities. This is especially the case with ENV RIs that are housed within the higher education sector, as opposed to public sector (government) operated programs.

We noted with interest, the author's observation that "the decision over how much to invest in an ENV RI development project depends on its present and expected economic value." Perhaps, despite the basic research focus for many ENV RIs, there is a tacit underpinning of anticipated economic return, in terms of improved forecasts or reduced societal risks. Another assertion that caught our attention was the "development of ENV RIs is mainly motivated by environmental risks and the need for improved observations and efficient forecast." Our first reaction was that this is not always the case – for example, with BGC Argo or Deep Argo, how much of the resulting data gathered translate directly into improved forecasts? We felt the question of dual use (basic and applied), could be expanded on in your paper. However, we recognize that estimating the potential economic benefits stemming from basic science research is likely much more difficult. This is an area where non-economic metrics (e.g. research impact, facility efficiency & reliability, society/community/policy impacts) may be come into play. But for the purpose of your paper, perhaps that part of the equation could be set aside. Since your focus is on forecasting and risk reduction benefits, we wondered if it may be useful to consider subdividing costs between research/education vs. operational applications for your cost-benefit analysis (CBA)?

We were also hoping to see something more from your CBA discussion. You introduce different methodologies and discuss some example applications in the marine sector. But we would have liked to see your recommendations of best approaches to CBA for the Argo example. In addition, an order-of-magnitude CBA for Argo would be of great interest, if within scope for this paper.

Specific Comments ——————————

P2, lines 13-17: Discussion focusses only on forecast uses of data, while data archival for long-term studies or applications is also an important responsibility for ENV RIs. Proper management of archival and data access systems requires significant personnel and technology investments.

P2, lines 18-21: Here is a place where you could expand on the discussion of "dual-use" nature of ENV-RIs.

Section 2.1.1: Please include Deep Argo in your list of Argo platforms. Also, you mention the initial objective and float increase rates, but we would like to see the current total count as well.

P3, line 10: In addition to the platforms you mention, there are in-situ observing systems, both autonomous and cabled.

P3, line 14: Satellite systems play 2 distinct roles: communications satellites acquire the telemetered data from ocean observing platforms, while environmental observing satellites acquire remote sensing data from the ocean.

P3, line 23: Would like to see a summary statement explaining that analyzing Argo's economics impacts in isolation does not capture the full up- and downstream benefits deriving from the combined applications of these different observing systems.

P4, line 14: We note that many geologic, habitat, marine chemistry, and biological data products are not developed for forecast-specific or operational audiences.

P3, line 24: You note that most suppliers or government owned or funded entities, it would be good to somehow indicate in Figure 5 which parts of the system are in which sectors.

P5, line 10: You mention Brazil and Canada – are these included in AtlantOS? Please revise sentence to explain more clearly.

Section 3.1.3: Costs reflect purchases only, but in our experience, there are also significant costs associated with maintenance (recovery, refurbishment, recalibration, redeployment). These generally exceed the initial purchase costs, but could perhaps be accounted for by the facility's operational budget.

P8, line 13: You discuss instrument inventories, however infrastructure costs are not included. In the case of some ENV RIs, infrastructure costs can greatly exceed instrumentation costs. But, perhaps this is not a significant cost factor for Argo.

Section 3.2.1: You introduce DAC/GDAC KPIs and complementary sources of KPIs. This could represent a study topic in itself (would be of interest to us), since different ENV RIs use different and sometimes widely varying KPIs. In the case of our facility, we focus on measurements to estimate research impact, facility efficiency & reliability, and societal impacts such as inter-organizational collaborations and influences on public policy.

P14: As mentioned above, we would like to see a stronger and more comprehensive conclusion to section 3. What is the take-home message of this section of your paper? Could you offer your assessment of the best approach to CBA for the Argo example? In addition, an order-of-magnitude CBA for Argo would be of great interest, if within scope for this paper.

Figure 1: Equipment supply industry is included, but not the significant contributions of the deployment and maintenance industry, including ships, remotely-operated vehicles, etc. Figure caption needs to be expanded – all figures must be stand-alone, so the

caption explains everything needed to understand the figure.

Figure 2: Figure caption needs to be expanded – all figures must be stand-alone, so the caption explains everything needed to understand the figure.

Figure 3: Figure caption needs to be expanded – all figures must be stand-alone, so the caption explains everything needed to understand the figure.

Figure 5: We find the diagram complicated to understand. The flow from Argo to DAC to end users is not very clear in the diagram. Would like to see the main flows emphasized. Is archive only at NODC, nowhere else? Also, as mentioned above, if sector (public/private) distinctions are important to your discussion, these should be indicated in the diagram. Which parts of these system incur costs or produce benefits that are accounted for in your CBA discussion?

Figure caption needs to be expanded – all figures must be stand-alone, so the caption explains everything needed to understand the figure. Thus, "Schematic representation of the Argo data flows from …. to …., including …..."

Table 1: There are a lot of blank cells in this table and we find it difficult to compare across areas. If US data are not available, it might be preferable to leave that column out entirely. We would like to see updated 2019 or 2018 economic data in this table. Suggested title: "Argo Programme Costs." Perhaps you could simplify this table by combining all categories (core, deep BCG) under each of the three areas. Could you extend the workforce FTE numbers across all three areas?

Table 2: We would like to see more up-to-date data if available.

Figure 6: Figure caption needs to be expanded – all figures must be stand-alone, so the caption explains everything needed to understand the figure.

Table 4: Suggested caption: "Four examples of cost benefit analyses of observing infrastructures used for ocean forecasting and climate monitoring." Please explain or spell out all acronyms.
Technical Corrections ──────────

There are numerous small grammatical errors in the text, for example the usage of "forecast" when "forecasting" or "forecasts" is preferred. Another example is the use of "scale" when "scales" is often preferred. We suggest you have the paper edited by a skilled English editor to eliminate these small errors and improve the readability.

P2, line 20: ". . .observations and accurate forecasts."

P2, line 25: ". . .observations at a global scale. It operates a growing array. . ."

P3, line 12: instruments are not just being mounted on marine mammals, so "animal-mounted instruments" instead of "sea mammal born."

P6, lines 3-4: ". . .instruments, airborne or waterborne platforms, data transmission technology, but also. . ."

P7, line 26: "Where information on. . ."

P8, line 8: ". . .Nation Centre for Scientific Research."

P8, line 30: ". . .latter; in such cases, the resolving power of the available information is critical to appraising the. . ."

P9, paragraph 1: "observational" data not observation data; "evaluate" performance rather than value

P10, line 7: ". . .and forecasting rely on. . ."

P10, line 14: "Forecasts are the outcome of . . .by comparing forecasts to natural runs."

P10, line 18: At "the global scale" or At "global scales".

P11, line 17: "While RMSD is used as a key statistic for. . ."

P 14, line 10: ". . .observation and forecasting is therefore considered beneficial. This conclusion is limited by the fact that the scenarios. . .
P14, lines 22-24: "Problematically, the two above conditions...CBAs are designed to meet immediate needs for making decisions..."

---

## Referee Comment (RC2) · Antti Pursula (Referee) · 6 Aug 2019

This paper reviews methods used for assessing economic impact of environmental research infrastructures. The topic is relevant for Geoscience Communication readers as the impact assessment of research investments is more and more in demand for decision-makers, and of interest to the society at large. The paper takes on to describe main tools used to analyze economic impacts together with their application scope and limits, reflecting them against the case study of Argo, an ocean observing system.

The methodology in the paper is sound and has the work has potential to be an interesting contribution. Slightly problematic part is the treatment of downstream impacts,

which are covered through a KPI analysis, not impact assessment methods. Regardless of this the manuscript gives valuable insights and information on its topic. However, I have several comments, listed below, that should be considered before publication.

1. I am missing more argumentation on some of the statements made in the manuscript (see section based notes below). There should be either references to the source, or more explanation in the text. This is especially the case in the Introduction section. One would expect to cite existing work at the beginning of the paper. For example OECD report ("Reference framework for assessing the scientific and socio-economic impact of research infrastructures", https://doi.org/10.1787/3ffee43b-en) has treated the topic quite extensively.

2. The classification of impacts to three categories seems reasonable. However, as these categories are used throughout the text, it should be explained why these categories were selected, and is this original contribution or adopted from literature. It would be useful to explain in more detail these categories in terms of what actors operate in each of them (e.g. suppliers, data processors, end-users etc). A figure showing relations of different aspects would be very good, and could replace or supplement the current Figures 1-3. This would also clarify especially the formation for downstream impacts. Also, perhaps "indirect impacts" is more descriptive than "feedback impacts" for the third category.

3. The presentation could be made more readable by clearly stating when discussing generally applicable parts (review of used methodologies in Environmental research infrastructures) and particulars of the case study (Argo observing system). Mixing these two is visible at least in section 3.2. Also, summary sections 3.1.4 and 3.2.3 can be improved by writing paragraph on findings applied to Argo case study. Also, I am missing a summary subsection on feedback / indirect impacts (under section 3.3).

4. The language used is sometimes confusing and not precise enough (see detailed notes below). A round of language checking would be beneficial for general readability.

**Detailed comments by section**

_Abstract:_

The abstract should be re-written. The first paragraph is unclear (for example, what are the "these activities" referred to. Should include a sentence on why this study is done, what is its value for readers.

The second paragraph on the objective of the paper is clearly explained but the text on case study (3rd paragraph) refers to understanding data flows which is not the central topic of this study.

_1. Introduction:_

This section needs improvements.

* 1st paragraph: Is this analysis of investments to ENV RIs; or ENV RI development projects? Should be used consistently.

* 2nd paragraph: Different objective for the paper as in the Abstract: "focus on the mechanism through which economic value is generated by ENV RI development projects". I think the objective written in the abstract is more suitable ("review of main tools . . .").

* 3rd paragraph: The presented classification is reasonable. Is it original contribution or is there a reference? Who are the actors in the Downstream impacts category and how are they different from the feedback (or indirect) impacts actors?

* Row 12 page 2: What is meant with "compensations" here?

* Final paragraph: Is there a reference for this statement: "The development of ENV RIs is mainly motivated by environmental risks and the need for improved observations and efficient forecast." This statement excludes for example improved preconditions for research & understanding the functioning of the earth system – that one thinks are relevant when investing in a Research infrastructure.
[Figure]

Adding short discussion on impact assessments on RIs performed earlier (with references) would be useful, for example between 2nd and 3rd paragraph.

_2. A case study: Argo_

This section would benefit from clarifications and references.

Subsection 2.1.1:

* Give a reference or web site for Argo and for GOOS.

* The first bullet point list would work better as normal text.

Subsection 2.1.2:

* "The latter provides real time in situ datasets to
Copernicus Marine Services and to ocean and climate scientists": Is this to understand only GDAC at Coriolis France provides this and the GDAC in USA does _not_ provide data to scientists?

* Page 4, row 3: "In Europe, the suppliers can be member state- and EC-funded entities". Can forecast suppliers not be private companies? That would be different than in weather forecasts, and would be worth to explain.

* Please provide reference or website to EMODNet published data products.

Subsection 2.1.3:

* Consider the title of the subsection. Perhaps "Development projects", or "Planned development of Argo system"

* The bullet point list on Remarks would work better as normal text.

* The first sentences, starting with "The present data market is recent. . ." make several claims concerning the marine data market, and would require a reference or explanations unless these are considered to be general knowledge.

* The text refers to figure 5 to illustrate supply chain, but the figure is about data flows.

Separate Figure on supply chain would be needed.

Subsection 2.2:

* Small comment that comparing staff costs in Europe should be compared to European part of the operating costs, not to the global operating costs.

_Section 3:_

This section is the main new contribution of the work and gives plenty of valuable information. However, it would benefit from clarifications, and in subsection 3.2 some changes are proposed.

_Subsection 3.1:_

In general the part on upstream impacts is well researched and written. Minor remarks:

* First row: "The focus is on the development . . . of ... development projects." The word 'development' is duplicated in this sentence.

Subsection 3.1.1:

Remark, not needing necessarily modifications in the text: Limitation, as I see, with this approach is that the cited study does not try to assess the impact of certain activity to MST companies. It merely scopes the size of the industry in general. This can provide good background information on further analysis.

Subsection 3.1.2:

* Page 7, row 3: "SBS are readily available". Please give a reference or web site.

* My impression is that this approach has the same limitation as the one in 3.1.1., that only background information about the total market size can be gained; and the impact of RIs is not possible to extract. Is this so? Perhaps this could be mentioned in the text.

Subsection 3.1.3:

* Page 7, row 26: "the latter permit to assess the direct impacts of a marginal demand increase on supply." Please elaborate on this point as it is a key point in this subsection. Does it mean simply that comparing RI investment to the size of the industry gives relevant importance of the RI based demand? Or is more sophisticated analysis proposed?

* In general, this approach to combine ENV RI purchases information & official statistics would be interesting to discuss further. If not in scope for the current paper, maybe for future work.

* Page 8, row 6: "It [Table 3] points out the high degree of competition on EU marine research equipment markets". I don't think such statement can be deduced from the Table 3. According to this table it could as well be that each instrument or purchase item is produced only by 1 single company, and there is no competition at all. Please reformulate.

_Subsection 3.2:_

The problem with this section is that it discusses key performance indicators of the activities or the RI. It is an important topic but the significance for the economic impacts of the RI is not presented. The OECD report referred to earlier discusses also this challenge.

The downstream impacts in a typical business case would be for the actors that refine RIs data outputs to create value added services. The argument in the paper is that there is no such relevant business at the moment. Would be interesting to list players in the downstream market and discuss this market in economic point of view. Can the growth of that sector be assessed in any way as a function of ENV RI data output? Why is there not such market at the moment? Can Argo case study be used?

I suggest considering to rewrite parts of this subsection, both to motivate the study of KPIs in this context and to include discussion of the potential economic impacts. Please

consider also shortening of the text. For example 3.2.2 is interesting information but perhaps partially out of scope.

* Page 9, row 4: "...assessment of downstream impacts requires valuing the performance of observation data products...". Are there tools for such evaluation ? This would be highly interesting.

* Page 9, row 11: Unclear sentence.

_Subsection 3.3:_

Overall this subsection is very valuable and interesting. Please consider adding a Summary subsection 3.3.3 (as in 3.1.4 and 3.2.3), with information on even qualitative CBA for the Argo infrastructure.

* 1st paragraph: "Feedback impacts (Figure 3) involve the entire chain of activities upstream and downstream of ENV RIs as a response to environmental risks and uncertainty". I find this definition limiting the scope in which ENV RI activities can create impacts. Most likely there are sectors outside environmental risk management that creates value based on ENV RI data. For example, travel industry, research services, optimization of operations and logistics, manufacturing of marine vessels, fishing, etc. In fact one of the cases discussed later (Gulf of Maine case) is not mainly on response to risks and uncertainty. I propose reformulating the mentioned sentence.

* 2nd paragraph: "...from improved forecast". Again, I feel that there is unnecessary limitation of scope. What about monitoring of the current state or even historical time series? Those can also be valuable to customers. This may be a minor thing but precise language is preferred.

* Page 12, row 28: Please explain acronym SAR

_4 Conclusions_

Good section would be still improved if it is possible to give some recommendations to
RIs, for example Argo, on what methods to use (or not use) at the moment for impact assessment.

_Figures and Tables:_

Figures 1 and 2 are difficult to grasp. The boxes contain different types of entities together. e.g. in Figure 2, Data processing is a function or process; Processed data supply is a market mechanism; and the rest are services. More informative would be figure in which different actors in the value chain are shown (equipment suppliers, RI, data processors, end users, industries benefiting from indirect impacts. . .). Captions of Figures 1-3 should be changed. They don't describe impacts per se.

---

## Author Comment (AC1) · 22 Aug 2019

Many thanks to referees for comments. I am preparing a new version of the paper.

---

## Author Comment (AC2) · 9 Nov 2019

Response to RC1

General comments

1. Comment 1

The main focus of the paper is on operational applications of ENV RIs. How can the paper better address basic research and education applications? How can the dual use nature of ENV RIs (basic research and operational uses) be better taken into account in the paper ? It would be useful to consider subdividing ENV RI costs between research

and education costs and operational costs.

Response to comment 1

The focus of the paper is on the economic impacts of ENV RIs through operational uses (monitoring and forecasting). But research and education applications are considered essential by the scientific community, in particular the Argo community. They must be given attention, first, by specifying, in the introduction and in the section on downstream impacts, the role of scientific research in the data supply chain ; second, by using information from the literature on RI impacts on research and education, and indicators used to quantify these impacts.

Regarding Argo specifically, the operational and research domains are complementary: Argo contributes to providing scientists with key data for research and education purposes, e.g. for ocean modelling and reanalysis. Ocean models are also used for monitoring and forecasting. E.g. scientists insist that Deep Argo will reduce the uncertainty in the rate of deep ocean warming, thus in future sea level rise projections.

Subdiving ENV RI costs between research costs and operational costs faces a difficulty. For an ENV RI such as Argo, research is not only a data user. Ocean models are developed by scientists, using observations from Argo and the other observing systems. Ocean models are then used both for research (ocean analysis) and for monitoring and forecasting. Research therefore contributes to operational uses. So ENV RI costs cannot be easily subdivided into costs for research and costs for operational applications.

Changes in manuscript

In the introduction and the downstream impact section, stress the significance of research and education as a user of ENV RI products. Mention quantitative indicators proposed in the literature to assess the impacts of RIs on research. In the description of Argo, mention the contribution of research to the development of marine services.

[Figure]

**2. Comment 2**

The CBA discussion lacks a stronger and more comprehensive conclusion and recommendations of the best approaches to CBA for Argo.

Response to comments 2

Conclusion on section 3 must be strengthened by building on a classification of the four examples of CBA, depending on their geographical scope and on the set of environmental information users involved. The CBA examples also differ with regard to the description of users' behaviour toward improved information (i.e. how they adapt their own business strategies to improved information). Classifying CBAs according to these features helps to address the question of CBA for Argo.

CBA for Argo : Argo is a component of the GOOS and contributes to improving ocean models. A CBA of Argo would therefore differ from the CBA examples given in the paper. The latter apply to the general impacts of entire observing systems, while the former would assess the impact of improved ocean information, starting from a baseline (no-Argo) scenario with an existing observing system. So the CBA examples assess the impact of the existence of an observing system, while a CBA for Argo would assess the impact of an improvement of an observing system.

Technical studies have been published on the use of improved ocean information, partly based on Argo observations, to support agriculture stocks and crop management, responses to oil spills and marine pollution, and search and rescue : the impacts that are described are not quantified in monetary terms. I failed to find any publication on a CBA of Argo specifically.

Changes in manuscript

After subsection 3.3.1, make a summary highlighting the main features of the CBA examples (spatial scale of observing systems under analysis, characteristics of data users). After the summary, add a new subsection to address the issue of CBA for

Argo.

Specific comments

Certain specific comments concern unclear explanations, sentences, figures or tables, and about missing elements. The best response is to make the appropriate corrections. Other specific comments (below) require responses and corrections.

3. Comment 3

Page 2, lines 13-17 and page 4, line 14 : the uses of data from ENV RIs are not limited to forecasting or operational applications but also include data archival for long-term studies.

Response to comment 3

Comment 3 is linked to comment 1 and refers to the research purposes of ENV RI data products. Indeed, the data from ENV RIs are a major source of information for research.

Changes in manuscript

In the introduction, mention data management and long-term studies as essential components to study the downstream impacts of ENV RIs.

4. Comment 4

Page 3, line 23 : analyzing Argo's economic impacts in isolation does not capture the full upstream and downstream benefits from the combined applications of observing systems.

Response to comment 4

A summary must be added and include this comment. Argo is used to illustrate the organization of the supply chain and give an order of magnitude of costs. Upstream, downstream and feedback impacts, as they are described in the paper, are not analyzed for Argo separately.

Changes in manuscript

In section 2.1, add a summary on Argo explaining its complementarity with other observing networks. Upstream and downstream impacts of ocean observing systems are therefore wider than those of Argo, taken separately.

5. Comment 5

Section 3.1.3 and Figure 1 : purchase costs are mentioned but maintenance costs are not. Figure 1 does not include contributions of the deployment and maintenance industry, including ships and remotely-operated vehicles, inter alia.

Response to comment 5

Indeed, it must be made clear that upstream impacts include deployment and maintenance costs. The Argo cost report referred to in section 2.2 includes these costs.

Changes in manuscript

The description of upstream impacts and the related figure have to be revised accordingly, as explained in the above response.

6. Comment 6

Page 8, line 13 : instrument inventories are discussed but infrastructure costs are not. These are high however.

Response to comment 6

It is true that infrastructure costs are not mentioned in the section on upstream impacts. Instrument inventories are used as an example of how purchases can be derived from available data on the stock of material. The objective is methodological.

Changes in manuscript
Mention infrastructure costs in the beginning of the section. Explain that instrument inventories (section 3.1.3), if available, are an example of how to obtain periodical purchase values.

7. Comment 7

Section 3.2.1 : KPIs are not limited to the small set mentioned in the paper. Different ENV RIs use very different KPIs, notably research specific KPIs.

Response to comment 7

It is indeed important to mention research related KPIs. Note that the KPIs discussed in subsection 3.2.1 are related to the performance of the observing system as a primary data acquisition system. Research, as a primary and processed data user, operates downstream of data acquisition and quality control. So we are not talking about the same segment of the supply chain. Research performance is partly driven by observing system performance.

Changes in manuscript

Mention the significance of research performance indicators in the introduction and in the section on downstream impacts.

8. Comment 8

Page 14 : section 3 lacks a stronger conclusion and recommendations on an approach to CBA for Argo.

Response to comment 8

See response to comment 2.

9. Comment 9

Page 3, line 24, and Figure 5 : indicate in Figure 5 which parts of the system are government owned or private. Indicate which parts of the public and private sectors

incur costs or produce benefits which are accounted for in the CBA discussion.

Response to comment 9

Figure 5 must be clarified and simplified. Public and private entities can be indicated. The examples of CBA used in section 3.3 discuss costs and benefits to society, in other terms : a) costs covered by public funding, so principally the costs incurred by public entities ; b) benefits to environment sensitive activities (private businesses or public coastal agencies and authorities). Figure 5 must give these indications ; section 3.3 must also clarify the cost-and-benefit question.

Changes in manuscript

Modify Figure 5 accordingly. Clarify the cost-and-benefit question in the beginning of section 3.3.

10. Comment 10

Tables 1 and 2 : simplify Table 1 and remove blank cells ; update Tables 1 and 2.

Response to comment 10

Table 1 must be simplified. The comparison between EU and US costs is difficult because of a dissymetry between certain countries in terms of available data. So the US column must be removed. Workforce number must also be removed as it concerns Euro-Argo only.

In terms of updating figures, Table 1 relies on a report published by the AtlantOs project (probably the only attempt to propose harmonised Argo cost figures for all participant countries). More up-to-date figures are included in the report but, given reliability issues, it is preferable not to use them.

Table 2 data would also need updating ; unfortunately, I failed to find an update of the study. However, this table is presented for methodological purposes : the idea is to show a type of survey which can be carried out to classify and analyze different

categories of RI suppliers.

Changes in manuscript

Simplify Table 1, remove the US column, change title.

11. Comment 11

Many English mistakes in the text.

Response to comment 11

The text is being edited. However a small number of expressions used by scientists cannot be changed, e.g. "nature run".

---

## Author Comment (AC3) · 9 Nov 2019

Response to RC2

General comments

1. Comment 1

Give more argumentation in several parts of the paper, in particular in the introduction. Cite existing works, e.g. the OECD report on the impacts of RIs.

Response to comment 1

[Figure]

It is necessary to strengthen the argumentation in several parts of the paper, in particular in the introduction. It is important to refer to the European Strategy Forum for Research Infrastructures (ESFRI) : its views motivated the ENVRI PLUS research project and the work summarized in the paper. It is also important to refer to OECD's analysis of RIs' impacts on science and economy. It must be noted that the field covered by the OECD report overlaps, but differs from, that of the paper. The OECD is especially useful to cover the impacts on research and education. Its methodology must be mentioned.

Changes in manuscript

In the introduction, explain more in details the motivations of the paper. Refer to the ESFRI Roadmap and to the OECD report on RI impacts and its methodology. Among the topics addressed by the ESFRI Roadmap, explain the focus on climate change and other environmental hazards, and the need for more knowledge in this area. In the downstream impact section, mention the OECD approach to assessing impacts on research.

2. Comment 2

Explain the reason for the selected classification of impacts. Explain the impacts in terms of categories of players. A figure would be a useful contribution. Clarify the formation of downstream impacts.

Response to comment 2

The best option is to explain the classification of impacts in the introduction. This classification is an original contribution, motivated by my own understanding of ENV RIs. The ESFRI Roadmap is also an important motivation. Figures 1 to 3 must be clarified or replaced by a more general figure including all impacts. It is feasible and useful to indicate the different categories of players. The formation of downstream impacts must be clarified in the introduction, in the beginning of section 3.2 and in a

figure.

Changes in manuscript

In the introduction, explain the motivations of the impact classification and refer to ESFRI. Modify and clarify figures 1 to 3 or replace them by another figure. Clarify the description of downstream impacts by indicating the different categories of actors.

3. Comment 3

Clarify the paper by indicating the generally applicable parts and the specific aspects of the case study. Improve sections 3.1.4 and 3.2.3 by a paragraph on findings applied to the case study. Add a summary on feedback impacts.

Response to comment 3

A clarification is necessary throughout all the text to distinguish between general features and Argo specific features. It is useful in particular to focus on the section summaries to clarify this.

Changes in manuscript

Revise sections 3.1.4 and 3.2.3 to clarify what is related to environmental impact assessment and to the Argo case study. Add a summary for 3.3.

4. Comment 4

The language is confusing and not precise enough. Language checking is necessary.

Response to comment 4

The text is being edited.

Detailed comments

5. Comment 5

Abstract. The abstract must be redone. Explain why the study is made and its value
for readers.

Response to comment 5

It is necessary to redo the abstract and clarify the text. It is necessary to explain the general context of the study, the importance of ENV RIs and the need for analyzing the economic aspects of their data supply chain. The need for reliable methods to do so must be stressed. Introduce Argo and its relevance as a case study.

Changes in manuscript

Abstract : modify abstract accordingly.

6. Comment 6

Improve introduction. Clarify the wording. Clarify the objectives. Specify the actors involved in downstream impacts. Clarify the context of the development of ENV RIs. Add a discussion on earlier RI assessment methods.

Response to comment 6

The introduction must be stengthened and the context of the study must be explained more in details. Reference must be made to the ESFRI Roadmap, in particular in relation to environmental issues. Reference must be made to the OECD approach to RIs' impacts. The motivations for the proposed impact classification must also be explained. It is however preferable to be brief about the actors involved in the downstream impacts and give details in section 3.2.

Changes in manuscript

Modify introduction accordingly.

7. Comment 7

Section 2 in general and section 2.1. Clarify section 2. Give reference. Clarify wording.

Response to comment 7
The Argo section must be improved, based on recent publications. This will not fundamentally change the description, but it is necessary to clarify some aspects of Argo and the supply chain. Figure 5 must be improved to better describe the supply chain.

Changes in manuscript

Modify section 2. Add literature references. Better describe the supply chain and the actors involved. Add a figure describing the supply chain.

8. Comment 8

Section 2.2. Staff cost comparison is not precise enough.

Response to comment 8

In order to have a homogeneous table and avoid blank cells, only Argo cost elements will remain in Table 1. So Euro-Argo specific costs must be removed.

9. Comment 9

Subsection 3.1.2. The SBS method has the same limitations as the Barrow study of section 3.1.1.

Response to comment 9

The ad-hoc inquiry-based method proposed by Barrow (section 3.1.1) is feasible if respondents accept to give business information. If yes, this permits to make a targeted analysis of the upstream supply companies. Barrow did it by limiting the business information that respondents had to give.

The structural business statistics give a considerable amount of business data per class of activity. The problem is that the activity classes are wide (i.e. each includes many enterprises) and not targeted enough ; the data which can be extracted are not very accurate for our purpose : assessing upstream impacts based on SBS is a second best method.

Changes in manuscript

More clearly explain the difference between the methods presented in 3.1.1 and 3.1.2.

10. Comment 10

Page 7, row 26. Clarify the explanation.

Response to comment 10

Using the CPA is valuable for data accuracy. The CPA includes more than 3000 categories (products and services) while the NACE has 615 classes. If RI investments are known in details and can be broken down by CPA category, it is feasible, in principle, to evaluate the impacts of investments in terms of value added increase, based on the branch accounts of the National Accounts. A condition is that branch accounts are available at class level.

Changes in manuscript

Give more detailed explanations in section 3.1.3 about the transition from the CPA to the National Accounts.

11. Comment 11

Page 8, row 6. Reformulate comments of Table 3.

Response to comment 11

Indeed, Table 3 does not inform about competition.

Changes in manuscript

Remove remarks on competition in comments on Table 3.

12. Comment 12

Section 3.2. Downstream impacts are covered through a KPI analysis, not impact assessment methods.

Response to comment 12

In the downstream impact section, as a first step, the paper focuses on the performance of ENV RIs in terms of environmental observations. This step is required for an approach to downstream impact assessment methods, because such performance is a major driver of the quality of research products and of the competitiveness of value-added services.

But it is true that such approach would miss the target if it included the above components only. Without losing sight of the main focus of this paper, the downstream impact section must discuss the methods used to assess impacts on research and education and on value-added service markets (value-added services include customized services for monitoring and predictions). The free-of-charge principle applies to the supply of observational data products and first stage monitoring services (Copernicus services) with suppliers being public entities. But value-added services may often be provided on competitive markets.

Note however that the indicators mentioned in the beginning of 3.2 partly contribute to a methodology for downstream impact assessment: the KPIs include, inter alia, "number of downloads" which measures the downstream demand for observational data. Further downstream, to focus on the case study, ocean models use data from the observing systems (after first stage processing), and help to increase knowledge on environment for different research fields and for operational purposes including monitoring and forecasting: this is partly measured by OSEs (presented after the KPIs), which are another component of downstream impact method. So, such indicators also concern the downstream impact assessment issue.

Changes in manuscript

Redraft and improve section 3.2 by adding a subsection on downstream impact assessment methods, covering impacts on research and education, and impacts on value-added service markets. In terms of Earth observation, Copernicus Marine Environ-
ment Monitoring Services (CMEMS) is a key component. The downstream markets generated by CMEMS must also be addressed in this subsection. Finally, raise the issue of correlations between the (first step) performance indicators and the downstream impact indicators addressed in the additional subsection.

13. Comment 13

Page 9, row 4. Are there tools for "valuing the performance of observation data products"?

Response to comment 13

Incorrect formulation. Read : "assessing the performance of data from ENV RIs". This assessment is made by the assembly centres which produce the KPIs mentioned in comment 12.

Changes in manuscript

Redraft beginning of 3.2.

14. Comment 14

Page 9, row 11. Unclear sentence.

Response to comment 14

The beginning of 3.2 must be revised and clarified. The different types of downstream impacts must be briefly outlined, including performance indicators and impacts on value-added markets. Then the Argo case study must be mentioned.

Changes in manuscript

Modify accordingly. Outline downstream impacts in the general case of ENV RIs. Then address the Argo case study in broad terms.

15. Comment 15

Subsection 3.3. Add a summary with information on CBA for Argo.

Response to comment 15

A summary is necessary for 3.3. It would be preferable to treat separately the lessons drawn from the CBA examples in a summary, and address the Argo issue in a separate subsection after the summary. This is because a methodological classification of the CBA examples can be presented in the summary and help to identify the main issues concerning an Argo CBA.

Changes in manuscript

Make three subsections in 3.3 : a) description of the CBA examples ; b) summary on the CBA examples ; c) the issue of feedback impacts from Argo.

16. Comment 16

Paragraph 1. The approach to feedback impacts in terms of environmental risks and uncertainty limits the scope in which ENV RI activities can create impacts.

Response to comment 16

It is true that the formulation is misleading and must be modified. The term of uncertainty was intended to express the idea that more ocean and weather knowledge is made available to players, so that their uncertainty (about the state of the environment) is reduced. But this term is confusing and must be changed.

Changes in manuscript

Modify paragraph accordingly, and insist on impacts in terms of knowledge increase.

17. Comment 17

Paragraph 2. The idea of improved forecast is a limitation of scope and ignores other beneficial objectives such as monitoring and archival.

Response to comment 17
It is true that the formulation is also misleading.

Changes in manuscript

Redraft accordingly, along the lines adopted for comment 16.

18. Comment 18

Page 12, row 28. What does SAR stand for ?

Response to comment 18

Search and rescue.

Changes in manuscript

Check and explain all acronyms.

19. Comment 19

Conclusion. Give recommendations on the treatment of RIs, e.g. Argo, in terms of methodology.

Response to comment 19

The conclusion must be improved, using the summaries on each type of impacts and the discussion of the Argo case in the feedback impact section. The main methodological issues must be emphasized, including data availability and accuracy ; assessment consistency, i.e. correlations between the metrics used for the different steps of the impact assessment ; and treatment of players' and data users' adaptation capacity to knowledge increase.

Changes in manuscript

Modify conclusion accordingly.

20. Comment 20

Figures 1 to 3. The figures are unclear and confusing. Limit description to actors. Change captions.

Response to comment 20

As said in comment 2, the figures must be changed or replaced by another one. Limiting the description to actors would help to avoid confusion.

Changes in manuscript

Modify figures accordingly.

––––––––––––––––––––––––––––

---

## Author Response (AR1)

Author's response

Assessing economic impacts of environmental research infrastructures : overview of methodolical tools (Regis Kalaydjian)

The point-by-point responses to reviewers were sent some weeks ago, and the changes in the manuscript follow the changes listed in the responses.

List of the changes :

- The abstract has been redrafted.
- The introduction has been redrafted.
- An introduction to section 2 has been added.
- The first paragraph of 2.1.1 has been redrafted.
- The summary on Argo (2.1.3) has been added.
- Section 3.1.4 has been clarified.
- Section 3.2 has been clarified in order to distinguish between comments on Argo and comment on ENV RIs, especially in the beginning of the subsections.
- 3.2.3 has been added to address more clearly the problem of correlations between indicators.
- 3.2.4 has been added to address the topic of research and education indicators.
- 3.2.5 has been slightly improved.
- 3.3.2 has been improved.
- 3.3.3 has been added to address the problem of specific feedback impacts from Argo.
- The conclusion has been redrafted.
- The captions of figures have been improved.
- Figure 6 has been removed. It did not provide clear information.

English language has been improved and the main mistakes have been corrected for the abstract, the introduction and section 1 and 2. Unfortunately, it was too late to correct section 3 and the conclusion. I beg the permission to make these remaining corrections.